# Novel Insights into miR-944 in Cancer

**DOI:** 10.3390/cancers14174232

**Published:** 2022-08-31

**Authors:** Jinze Shen, Qurui Wang, Chenhao Liang, Xinming Su, Yufei Ke, Yunan Mao, Jie Fang, Shiwei Duan

**Affiliations:** Department of Clinical Medicine, Zhejiang University City College School of Medicine, Hangzhou 310015, China

**Keywords:** miR-944, ceRNA, dysregulation, diagnosis, prognosis, drug

## Abstract

**Simple Summary:**

miR-944 is localized in intron 4 of TP63. ΔNp63 in intron 3 of TP63 recruits the transcription factor AP-2 to promote miR-944 gene expression, which mediates epidermal differentiation induction by ΔNp63. miR-944 is dysregulated in various cancers. In squamous cell carcinoma. miR-944 can target and inhibit 27 protein-coding genes, thereby regulating cell cycle, proliferation, apoptosis, epithelial mesenchymal transition, cancer cell invasion and migration, and other cell behaviors. The genes targeted by miR-944 are involved in three signaling pathways, including the Wnt/β-catenin pathway, Jak/STAT3 pathway, and PI3K/AKT pathway. miR-944 was regulated by a total of 11 competing endogenous RNAs, including 6 circular RNAs and 5 long non-coding RNAs. Abnormally expressed miR-944 can act as an independent prognostic factor and is closely related to tumor invasion, lymph node metastasis, TNM staging, and drug resistance. miR-944 is expected to become a critical biomarker with great clinical application value in cancer.

**Abstract:**

miRNA is a class of endogenous short-chain non-coding RNAs consisting of about 22 nucleotides. miR-944 is located in the fourth intron of the TP63 gene in the 3q28 region. miR-944 is abnormally expressed in cancers in multiple systems including neural, endocrine, respiratory, reproductive, and digestive systems. miR-944 can target at least 27 protein-coding genes. miR-944 can regulate a series of cell behaviors, such as cell cycle, proliferation, invasion and migration, EMT, apoptosis, etc. miR-944 participates in the networks of 11 ceRNAs, including six circRNAs and five lncRNAs. miR-944 is involved in three signaling pathways. The abnormal expression of miR-944 is closely related to the clinicopathological conditions of various cancer patients. Deregulated expression of miR-944 is significantly associated with clinicopathology and prognosis in cancer patients. In addition, miR-944 is also associated with the development of DDP, RAPA, DOX, and PTX resistance in cancer cells. miR-944 is involved in the anticancer molecular mechanisms of matrine and Rhenium-liposome drugs. In conclusion, this work systematically summarizes the related findings of miR-944, which will provide potential hints for follow-up research on miR-944.

## 1. Introduction

microRNAs (miRNAs) are endogenous short non-coding RNAs of approximately 22 nt that typically target the 3’ untranslated region (3’-UTR) of mRNAs [1], thereby inhibiting the function of protein-coding genes [2]. Dysregulation of miRNAs is often associated with the malignant transformation of cells, thereby participating in biological processes that promote cancer progression, metastasis, and treatment resistance [3,4].

miR-944 is located in the fourth intron of tumor protein p63 (TP63) in the chromosome 3q28 region [5] and produced at the 3’ end of the stem-loop structure of pre-mir-944. miR-944 is aberrantly expressed in more than 10 cancers. Targeted inhibition of mRNA by miRNA can be hindered by competing endogenous RNAs (ceRNAs), such as circular RNAs (circRNAs) and long non-coding RNAs (lncRNAs) [6]. miR-944 is regulated by eleven ceRNAs, including six circRNAs and five lncRNAs. miR-944 can target and suppress 27 protein-coding genes, thereby regulating cancer cell behaviors such as cancer cell cycle, growth, proliferation, epithelial-mesenchymal transition (EMT), cancer cell invasion, and metastasis. The genes targeted by miR-944 are involved in three signaling pathways, including the Wnt/β-catenin pathway, the Jak/STAT3 pathway, and the PI3K/AKT pathway.

In patients with nasopharyngeal carcinoma (NPC) [7], colorectal cancer (CRC) [8,9], or breast cancer (BrC) [10], low expression of miR-944 was not only associated with poorer overall survival (OS) but also with more advanced tumor infiltration, more lymph node metastasis, and more advanced tumor node metastasis (TNM) stage [9]. miR-944 is associated with resistance to four anticancer drugs, including cisplatin (DDP) [11], rapamycin (RAPA) [12], doxorubicin (DOX) [13], and paclitaxel (PTX) [14]. miR-944 is involved in the molecular mechanism of action of two anticancer drugs, including the quinazolidine alkaloid matrine [15] and a drug under clinical trials, 188Re-liposome [16].

More and more studies have confirmed that miR-944 plays an important role in tumorigenesis and development. Since there is no systematic summary of miR-944, we review the research progress of abnormal expression, molecular mechanism, and clinical significance of miR-944. Our work aims to provide hints for future research related to miR-944.

## 2. miR-944 and Its Host Gene TP63

The p63 transcription factor encoded by TP63, the host gene of miR-944, is a tumor suppressor gene belonging to the p53 family [17]. N-terminal truncated isoform of p63 (ΔNp63), which is transcribed from an alternative promoter in intron 3 of TP63, can regulate the epithelial properties of cells and play an important role in the terminal differentiation and stemness maintenance of basal epidermal cells [18]. miR-944 was significantly positively correlated with ΔNp63 expression in cervical cancer (CxCa) and jointly exerted a cancer-promoting effect [19]. During the differentiation of human epidermal keratinocytes, ΔNp63 can recruit the transcription factor AP-2 to the promoter region of the miR-944 gene, thereby promoting the expression of miR-944 and inducing epidermal differentiation [17].

## 3. Aberrant Expression of miR-944 in Cancer

As shown in Table 1, miR-944 is downregulated in cells and tissues of 11 cancers. Meanwhile, miR-944 is lowly expressed in the plasma of esophageal cancer (ECa) [20]. It is worth noting that miR-944 is highly expressed in cancer tissues of LUSC [21] and endometrial carcinoma (EC) [22], as well as CxCa cell lines, serum, and tissues [5,23,24,25]. Furthermore, the expression of miR-944 in BrC is controversial. Specifically, miR-944 was highly expressed in serum and tissues of BrC [26], whereas miR-944 was found to be downregulated in five BrC cell lines and tissues [27].

We downloaded the TCGA (pan-cancer) dataset from the UCSC Xena database (https://xenabrowser.net (accessed on 20 June 2022)). We extracted miR-944 expression data (RPM) in 33 cancer samples and performed log2(RPM+1) transformation. We excluded cancer types with <3 control samples and finally retrieved miR-944 expression data in 16 TCGA cancer types. We calculated the quantile percentage of miR-944 expression among all non-zero-expressed miRNAs in each of these 16 cancer types. As shown in Figure 1A, miR-944 was highly expressed in nine tumors (0.75–1.0 quantile, Q4), while miR-944 was moderately expressed in other seven tumors (0.5–0.75 quantile, Q3).

We compared differences in miR-944 expression between non-tumor and tumor samples of 16 cancer types (unpaired Wilcoxon test, R version 4.1.3). As shown in Table 2 and Figure 1B, We found that miR-944 was significantly upregulated in five tumors (BLCA, HNSC, LUSC, THCA, and UCEC); significantly downregulated in two tumors (BRCA and PRAD) (Figure 1B). Notably, our TCGA analysis demonstrated the association of miR-944 expression with cancer risk in bladder cancer, head and neck squamous cell carcinoma (HNSCC), thyroid cancer, and prostate cancer, which has not been reported yet.

We also calculated differences in the expression of miR-944 between patients of different genders or races based on the TCGA database. The results were shown in Appendix A. There was no significant difference in the expression level of miR-944 between males and females in cancer. In BLCA and ESCA, the level of miR-944 in whites was significantly lower than that in other races.

## 4. Co-Expression of TP63 Transcripts and miR-944

miR-944 is located in the TP63. Previous studies have shown that ΔNp63, but not TAp63, can directly regulate the expression level of miR-944 by recruiting the transcription factor AP-2 [19]. The high correlation between the expression of TAp63 and miR-944 may be due to the common upstream regulators of TAp63 and ΔNp63, resulting in a significant positive correlation between TAp63 and ΔNp63.

In order to explore the relationship between TP63 and miR-944, we obtained the expression data of miR-944, TAp63, and ΔNp63 in TCGA (pan-cancer) from the UCSC Xena database (https://xenabrowser.net (accessed on 20 June 2022)).

Among 30 cancer types, we calculated pairwise correlations between miR-944, TP63, and ΔNp63 (Pearson’s correlation test). As shown in Figure 2, miR-944 expression was significantly positively correlated with TAp63 and ΔNp63 in 15 and 16 cancers, respectively (*p* < 0.01 and r > 0.5). In ACC, CHOL, OV, PCPG, and READ, the expression level of miR-944 was not significantly correlated with TAp63 and ΔNp63 (*p* > 0.05 or r < 0.3). Among them, the number of ACC (*n* = 19) and CHOL (*n* = 20) samples is small, which may lead to the above insignificant correlation. In KIRC and KICH, the expression of miR-944 was not significantly correlated with ΔNp63 (*p* > 0.05 or r < 0.3) but had a positive correlation with TAp63 (*p* < 0.01 and r > 0.45), suggesting that there may be a different regulatory mechanism of miR-944 expression in KIRC and KICH.

## 5. TP63 SNV and miR-944 Expression

Single nucleotide variant (SNV) occurs by a single base change caused by base substitution, single base insertion, or base deletion, including missense mutation, nonsense mutation, frameshift mutation, etc. [40]. SNV data for the TP63 gene in TCGA samples were downloaded from USCS Xena (https://xenabrowser.net/ (accessed on 20 June 2022)). In the TCGA samples, most TP63 mutations were missense mutations, and the primary mutation type was SNV (Figure 1C). TP63 mutation rates were higher in CESC, STAD, UCEC, SKCM, and BLCA (>3.0%, Figure 1D), but there was no significant association between TP63 SNVs and miR-944 expression (Figure 1E).

## 6. TP63 Copy Number Variation (CNV) and miR-944

CNV is a chromosomal structural variation, often caused by genome rearrangement, resulting in duplication, deletion, or copy number change in specific regions of the genome [41]. We downloaded the CNV of the TP63 gene from UCSC Xena (https://xenabrowser.net/ (accessed on 20 June 2022)). We calculated the correlation of TP63 CNV with TP63, ΔNp63, and miR-944 expression (Pearson’s correlation test). As shown in Figure 2, in CESC, ESCA, LUSC, PAAD, and TGCT, the expression level of miR-944 was significantly positively correlated with TP63 CNV (*p* < 0.0001 and r > 0.3). Notably, in cancers with significantly upregulated miR-944 (BLCA, HNSC, and LUSC), the expression level of miR-944 was at Q4, higher than that of most miRNAs, and miR-944 expression was positively correlated with TP63 CNV. However, in BRCA and PRAD where miR-944 was significantly downregulated, miR-944 expression was not significantly correlated with TP63 CNV.

## 7. miR-944 and Cancer Cell Behaviors

As shown in Table 3 and Figure 3, miR-944 can inhibit 27 protein-coding genes, thereby regulating various behaviors of cancer cells, such as cell cycle, proliferation, EMT, apoptosis, invasion, and migration.

## 8. Regulatory Effect of miR-944 on Cell Cycle

The cell cycle, a continual event in which a cell replicates its genetic material, grows, and divides into two daughter cells, is a ubiquitous and tightly regulated process [46]. Checkpoints at various stages of the cell cycle transmit abnormal signals to effectors that trigger cell cycle arrest [47].

In non-small-cell lung cancer (NSCLC), the CircFUT8/miR-944/YES1 axis can increase the proportion of cells in the G0/G1 phase of H522 and H1975 cell lines, reduce the proportion of cells in the S phase, and inhibit the cell cycle [15]. In CRC, miR-944 can block cell cycle G1 phase progression in the HCT116 cell line by targeting COP1 and MDM2 [37]. In EC, miR-944 downregulated CADM2 to promote cell cycle progression in two EC cell lines (Ishikawa and KLE), while miR-944 knockdown resulted in cell cycle arrest in the G1 phase [22].

## 9. Regulatory Effect of miR-944 on Cancer Cell Proliferation

Cell proliferation is tightly regulated in normal organisms, and uncontrolled cell proliferation can lead to cancer [48]. miR-944 targets eight genes, including VEGFC [28], STAT1 [30], MACC1 [9,29,35], EPHA7 [42], COP1 [37], MDM2 [37], GATA6 [38], and VEGF [39]. The inhibition of proliferation of various tumor cells involves 10 miR-944-related signaling axes, including CircBACH2/miR-944/HNRNPC [10], CircCSPP1/miR-944/FZD7 [13], CircHAS2/miR-944/PPM1E [34], CircSERPINA3/miR-944/MDM2 [7], CircZFR/miR-944/LASP1 [11], JPX/miR-944/CDH2 [43], PRNCR1/miR-944/HOXB5 [33], SNHG6/miR-944/ETS1 [32], SNHG6/miR-944/RAB11A [45], and CircFUT8/miR-944/YES1 [15]. In contrast, miR-944 promotes the corresponding cancer cell proliferation by targeting three genes (SOCS4 [21], CADM2 [22], and HECW2 [25]) and participating in the LINC00899/miR-944/ESR1signaling axis [23].

## 10. Regulatory Effect of miR-944 on Cancer Cell Apoptosis

Apoptosis is an important mechanism for controlling cell proliferation, maintaining tissue homeostasis, and eliminating harmful or unnecessary cells (e.g., cancer cells) in multicellular organisms [49]. miR-944 can participate in three signaling axes to promote cancer cell apoptosis, including the PRNCR1/miR-944/HOXB5 [33], CircCSPP1/miR-944/FZD7 [13], and CircFUT8/miR-944/YES1 [15] signaling axes. Notably, miR-944 was also able to inhibit cancer cell apoptosis by targeting CADM2 [22].

## 11. Regulatory Effect of miR-944 on EMT

EMT is a biological process in which epithelial cells acquire mesenchymal characteristics. EMT is closely related to the occurrence, development, and recurrence of tumors [50]. miR-944 inhibits EMT progression in cancer cells by targeting two genes (MACC1 [29,35] and GATA6 [38]). miR-944 can also inhibit the EMT process of cancer cells by participating in the SNHG6/miR-944/RAB11A axis [45].

## 12. Regulatory Effect of miR-944 on Cancer Cell Invasion and Migration

The invasion of tumor cells into surrounding tissues and their migration in blood vessels is an important initial step in tumor metastasis and is the main reason for the high incidence and mortality of cancer [51]. miR-944 inhibits cancer cell invasion and migration by targeting six genes and ten related signaling axes. The six target genes of miR-944 are VEGFC [28], MACC1 [9,29,35], GATA6 [8,38], VEGF [39], SIAH1 [27], and PTP4A1 [27]. The 10 miR-944-related signaling axes include the FGD5-AS1/miR-944/MACC1 [31], SNHG6/miR-944/ETS1 [32], PRNCR1/miR-944/HOXB5 [33], CircSERPINA3/miR-944/MDM2 [7], JPX/miR-944/CDH2 [43], SNHG6/miR-944/RAB11A [45], CircCSPP1/miR-944/FZD7 [13], CircFUT8/miR-944/YES1 [15], CircHAS2/miR-944/PPM1E [34], and CircZFR/miR-944/LASP1 [11].

In contrast to the above, miR-944 can also promote the invasion and migration of cancer cells by targeting three genes (CISH [44], SOCS4 [21], and HECW2 [25]) and the LINC00899/miR-944/ESR1 signaling axis [23].

## 13. The ceRNA Network of miR-944 in Cancer

CircRNAs are a new class of regulatory RNAs with a covalent closed-loop structure [52]. Long non-coding RNA (lncRNA) is a non-coding RNA with a length of more than 200 nt that can regulate transcription, epigenetic modification, and translation [53]. CircRNAs, lncRNAs, and miRNAs constitute a ceRNA regulatory network to regulate downstream protein-coding gene expression [54]. As shown in Table 4 and Figure 4, miR-944 can constitute 11 ceRNA regulatory axes, involving six circRNAs (CircFUT8, CircHAS2, CircCSPP1, CircSERPINA3, CircBACH2, and CircZFR) and five lncRNAs (JPX, SNHG6, LINC00899, PRNCR1, and FGD5-AS1).

In NSCLC, miR-944 is involved in four ceRNA signaling axes. The circFUT8/miR-944/YES1 axis [15], the FGD5-AS1/miR-944/MACC1 axis [31], and the SNHG6/miR-944/ETS1 axis [32] can promote the cell cycle, proliferation, invasion, and migration. The circFUT8/miR-944/YES1 axis can inhibit the apoptosis of NSCLC cancer cells and promote tumor growth in the H522 cell xenograft nude mice model [15]. The CircZFR/miR-944/LASP1 axis promotes the malignant phenotype of cancer and induces DDP resistance in cancer cells [11]. In NPC, the circSERPINA3/miR-944/MDM2 axis promotes the proliferation and invasion of NPC cells [7]. In gastric cancer (GC), the CircHAS2/miR-944/PPM1E axis promotes the proliferation, invasion, and migration of GC cells [34]. In CRC, the circCSPP1/miR-944/FZD7 axis promotes CRC cell proliferation, migration, and invasion, and inhibits apoptosis and DOX resistance. At the same time, the CircCSPP1/miR-944/FZD7 axis can also sensitize tumors to DOX in the LoVo xenograft nude mice [13]. In BrC, the circBACH2/miR-944/HNRNPC axis promotes the proliferation of tumor cells [10]. In tongue squamous cell carcinoma (TSCC), the PRNCR1/miR-944/HOXB5 axis promotes the tumor cell proliferation, invasion, and migration, and inhibits apoptosis, while the PRNCR1/miR-944/HOXB5 axis can also promote tumor growth in the SCC-9 xenograft nude mice [33]. In oral squamous cell carcinoma (OSCC), the JPX/miR-944/CDH2 axis promotes the proliferation, invasion, and migration of OSCC cells [43]. In pituitary adenoma (PA), the SNHG6/miR-944/RAB11A axis promotes cancer cell proliferation, invasion, migration, and EMT [45]. In CxCa, the CircZFR/miR-944/IL-10 axis promotes cancer cell resistance to PTX [14]. Meanwhile, the LINC00899/miR-944/ESR1 axis can also inhibit the proliferation, invasion, and migration of CxCa cells [23].

## 14. Exogenous Regulators of miR-944

As shown in Figure 5B, studies have shown that exogenous factors such as 4-(methylnitrosamino)-1-(3-pyridyl)-1-butanone (NNK), acetaldehyde, alcohol, and human papillomavirus (HPV) can affect the expression of miR-944 in vivo and play a role in the occurrence and development of cancer.

NNK is the main component of tobacco extract [44]. CISH contains an SH2 domain and a SOCS box domain that endogenously represses STAT [44]. NNK upregulates the expression level of miR-944 in OSCC cell lines (OEC-M1 and SCC-25), thereby increasing the targeted inhibition of CISH by miR-944, which in turn impedes the activation of the Jak/STAT3 signaling pathway and promotes Inflammation [44]. Acetaldehyde and alcohol can upregulate the expression of miR-944 in HCC, which in turn increases the sensitivity of HCC cells to doxorubicin [36].

We also calculated the effect of tobacco and alcohol history on the expression level of miR-944 in patients based on the TCGA database. The results were shown in Appendix A. In LUAD, the expression level of miR-944 in light smoking patients was significantly lower than that in non-smoking patients; however, drinking history did not affect the expression level of miR-944 in patients.

HPV is the leading cause of CxCa, which is the fourth leading cause of death in women worldwide [55]. HPV E6/E7 can significantly upregulate the expression level of miR-944 in HPV-infected CxCa tissues, resulting in larger tumors, later international federation of gynecology and obstetrics (FIGO) stage, and a higher level of lymph node metastasis rate [5].

## 15. miR-944 Is Involved in a Variety of Cancer-Related Signaling Pathways

As shown in Figure 6, miR-944 is involved in three signaling pathways that regulate the occurrence and development of cancer. miR-944 is involved in regulating the Wnt/β-catenin signaling pathway by targeting GATA6 in CRC [8]. miR-944 is involved in regulating the Jak/STAT3 signaling pathway by targeting CISH in OSCC [44], STAT1 in LUAD [30], and SOCS4 in LUSC [21]. In the PI3K/AKT signaling pathway, miR-944 targets VEGF in osteosarcoma (SaOS) [39], MACC1 in GC [35], and MDM2 in NPC [7] and CRC [37].

## 16. miR-944 and the Wnt/β-Catenin Pathway

The Wnt/β-catenin signaling pathway drives tumor cell proliferation, EMT, invasion, and migration, and plays an important role in tumor recurrence and metastasis [56].

GATA6 can promote the accumulation of β-catenin in the cytoplasm by enhancing EGF signaling or increasing the level of intracytoplasmic calcium [8]. After entering the nucleus, free β-catenin binds to transcription factors of the TCF/LEF family, activates the Wnt/β-catenin pathway, and initiates transcription of downstream genes [57]. miR-944 inhibits the activation of the Wnt/β-catenin signaling pathway by targeting GATA6, thereby inhibiting the EGF-induced EMT process in CRC cancer cell lines (HCT116 and SW480) [8].

## 17. miR-944 and the Jak/STAT Pathway

The Jak/STAT signaling pathway is involved in many biological processes that promote tumor cell proliferation, survival, invasion, and migration [58]. The Jak/STAT signaling pathway is overactivated in most cancers and is often associated with poor clinical prognosis [58].

STAT1 and STAT3 are important members of the STAT family, both of which can promote cell survival and induce immune tolerance [59]. CISH is a negative regulator of endogenous STAT3 signaling [44]. In OSCC, miR-944 can target and inhibit the expression level of CISH, promote the activation of the Jak/STAT signaling pathway, and then promote the occurrence of inflammation and the invasion and migration of cancer cells [44]. In LUAD, miR-944 directly targets and inhibits STAT1 expression, inhibits the activation of the Jak/STAT signaling pathway, thereby inhibiting cancer cell proliferation, and also hinders tumor growth in BALB/c nude mice [30].

SOCS4 can inactivate Jak, thereby blocking the activation of the Jak/STAT signaling pathway [60]. In LUSC, miR-944 targets and inhibits SOCS4, which in turn activates the Jak/STAT signaling pathway and promotes the growth and proliferation of cancer cells [21].

## 18. miR-944 and the PI3K/AKT Pathway

The PI3K/AKT signaling pathway can regulate a variety of key epigenetic modifiers and promote the occurrence and development of cancer [61]. Abnormal activation of the PI3K/AKT signaling pathway can promote the proliferation and metastasis of cancer cells and can also promote angiogenesis and induce drug resistance in cancer cells [62].

VEGF can activate Src and PI3K in turn by binding to membrane receptors [63]. PI3K promotes the phosphorylation of PIP2 to generate PIP3, which recruits and activates AKT at the plasma membrane, thereby activating the PI3K/AKT signaling pathway [64]. In SaOS, low expression of miR-944 can upregulate VEGF expression, activate PI3K/AKT signaling pathway, and promote cancer cell proliferation [39].

MACC1 can activate the PI3K/AKT signaling pathway by increasing c-MET levels [65]. In GC, under-expressed miR-944 upregulates MACC1 expression and activates the PI3K/AKT signaling pathway to promote cancer cell invasion and migration [35].

MDM2 inhibits the activity of the transcription factor p53 and degrades it by ubiquitination [66]. In NPC [7] and CRC [37], low expression of miR-944 upregulates MDM2 expression, inhibits p53 transcriptional activity, and promotes the cell cycle of cancer cells.

## 19. Prognostic Value of miR-944

The dysregulation of miR-944 is not only closely related to the pathological status of cancer tissues but also significantly related to the diagnosis of cancer risk and the prognosis of patients. As shown in Table 5, in CRC, HCC, NPC, and BrC, low expression of miR-944 was closely associated with poor patient prognosis. In CxCa, low expression of miR-944 was associated with better patient prognosis.

In NPC, low expression of miR-944 was associated with advanced clinical stage and was significantly associated with shorter OS [7]. In CRC, low expression of miR-944 was associated with advanced tumor invasion stage, lymph node and distant metastasis stage, TNM stage, more liver metastasis, and shorter OS and progression-free survival (PFS) [8,9,37]. In BrC, low expression of miR-944 was associated with advanced clinical-stage, late TNM stage, and shorter OS [10,67].

In LUAD, STAT1 is a downstream target of miR-944, and STAT1 expression is negatively correlated with miR-944 and significantly associated with shorter OS [30]. In addition, ceRNAs of miR-944 were highly expressed in four cancers including NPC, GC, NSCLC, and TSCC, and they could significantly suppress miR-944 expression and were associated with a poorer prognosis in cancer patients. In NPC, highly expressed circSERPINA3 inhibited the miR-944 expression and was significantly associated with later clinical stage and shorter OS [7]. In GC, highly expressed circHAS2 was negatively correlated with miR-944 and significantly correlated with later T stage, later TNM stage, more aggressive lymph node metastasis and neural/vascular invasion, and shorter OS [34]. In NSCLC, high expression of SNHG6 was negatively correlated with miR-944 and was significantly associated with later TNM stage, larger tumor size, and shorter OS [32]. In TSCC, high expression of PRNCR1 was negatively correlated with miR-944 and was significantly associated with larger tumor size, later clinical stage, more lymph node metastasis, and shorter OS [33].

We used CancerMIRNome (http://bioinfo.jialab-ucr.org/CancerMIRNome/ (accessed on 25 August 2022)) to explore the difference in the prognosis of patients with high and low expression of miR-944 in the TCGA database (PMID: 34500460). The results were shown in Appendix A. In BLCA, patients with high expression of miR-944 had better OS, while patients with high expression of miR-944 in LGG and THYM had poor prognoses (*p* < 0.05).

## 20. miR-944 and Its Therapeutic Value

Studies have shown that miRNAs are promising as novel tools or therapeutic targets for cancer therapy [68]. In five cancers (BrC, NSCLC, T-cell acute lymphoblastic leukemia (T-ALL), CRC, and CxCa), miR-944 was significantly associated with drug resistance in cancer cells. In NSCLC and HNSCC, miR-944 is involved in the molecular mechanism of action of two anticancer drugs.

## 21. miR-944 and Drug Resistance of Cancer Cells

Drug resistance of cancer cells is one of the important reasons for chemotherapy failure [69]. As shown in Figure 7A, miR-944 is involved in the resistance of cancer cells to various anticancer drugs.

DDP is a platinum-based anticancer drug widely used in the treatment of various solid cancers [70]. DDP induces apoptosis by interacting with purine bases on DNA to generate DNA damage [71]. In BrC, highly expressed miR-944 inhibits the BINP3/MMP/caspase3 axis, resulting in DDP resistance in MCF-7 cells [26]. In NSCLC, high expression of CircZFR activates the CircZFR/miR-944/LASP1 axis, resulting in DDP resistance in A549 and H1299 cell lines [11].

RAPA is a macrolide that specifically inhibits mTOR [72,73]. The mTOR signaling pathway is a major regulator of cell growth and metabolism, and its abnormal expression can induce a variety of human diseases, such as cancer, diabetes, and neurological diseases [74]. In T-ALL, miR-944 inhibits the THBS1/MMP2 axis, thereby inducing resistance to RAPA in Molt-4 cells [12].

DOX is an anthracycline antibiotic commonly used in the treatment of breast cancer, Hodgkin’s disease, and lymphoblastic leukemia [75]. DOX can induce apoptosis of cancer cells by regulating cell autophagy, but it also has certain toxicity to other organs, such as the heart [76]. In CRC, activation of the CircCSPP1/miR-944/FZD7 axis leads to elevated levels of downstream *p*-gp, LRP, and MRP, which in turn lead to resistance to DOX in LoVo and HCT116 cells [13].

PTX, a member of the paclitaxel family, can block mitosis and cause apoptosis [77]. IL10 has immunosuppressive and tumor-promoting abilities [78]. In CxCa, activation of the CircZFR/miR-944/IL10 axis leads to resistance to PTX in HeLa and SiHa cells [14].

## 22. miR-944 and Drug Therapy

As shown in Figure 5A, miR-944 was associated with the molecular mechanism of two anticancer drugs (matrine and 188Re-liposome).

Matrine is a quinazolidine alkaloid isolated from the roots of Sophora flavescens and Sophora japonica [79]. Matrine and its derivatives have anti-cancer, anti-inflammatory, analgesic, and anti-fibrotic functions [79]. In NSCLC, matrine inhibits CircFUT8/miR-944/YES1 axis, which in turn hinders cancer cell migration, invasion, and cell cycle, and promotes apoptosis [15].

188Re-liposome is a radioactive nanoparticle that emits γ and β rays, which can be used for imaging validation and cancer therapy, respectively. 188Re-liposome can be preferentially distributed in tumors, and its anticancer effect is currently in clinical trials [80]. In HNSCC, 188Re-liposome can downregulate miR-944, thereby inhibiting tumor growth [16].

## 23. The PCGs of miR-944 and Their Targeted Drugs

As shown in Figure 7B, by searching in the CADDIE database (https://exbio.wzw.tum.de/caddie/ (accessed on 15 July 2022)) [81], we found that multiple downstream protein-coding genes (PCGs) of miR-944 had targeted drugs, including CDH2 (Methadone), EPHA7 (Gefitinib, Sorafenib, Erlotinib, Imatinib, and Dasatinib), ESR1(Masoprocol, Diethylstilbestrol, Conjugated estrogens, Etonogestrel, and Desogestrel), IL10 (Efavirenz), MDM2 (Apomorphine, Cytarabine, Zinc, Thiram, and Zinc acetate), VEGF (Aflibercept), and YES1 (Gefitinib, Sorafenib, Erlotinib, Imatinib, and Dasatinib). In the future, it is necessary to explore whether miR-944 may interact with these drugs.

## 24. Discussion

miR-944 is located in the fourth intron of TP63. An in-depth study of the relationship between miR-944 and TP63 and ΔNp63 can broaden the understanding of the molecular mechanism of miR-944 in cancer. In the current study, the abnormal expression pattern of miR-944 in most cancers has been roughly clear. However, further study of the abnormal expression patterns of miR-944 in different BrC subtypes is required in BrC. Furthermore, we can infer that TP63 SNV has no significant effect on the expression level of miR-944, while TP63 CNV has a significant effect on the expression level of miR-944, mainly in squamous cell carcinoma.

The expression of miR-944 levels in BrC is still controversial. It was found that the expression level of miR-944 was lower in BrC cell lines MDA-MB-231, MCF-7, MDA-MB-453, ZR-75, and T47-D than in non-cancer cell line MCF-10A. The expression of miR-944 was lower in locally invasive breast tumor tissues from BrC patients than in adjacent tissues [27]. However, some studies have found that the level of miR-944 in cancer tissues and serum of BrC patients is upregulated compared with the serum of adjacent tissues and normal people [26]. These differences may arise from tissue heterogeneity. Further in-depth study of the expression levels of miR-944 in different cancer subtypes is required. Previous TCGA database analysis used the optimal cut-off method of miR-944 expression level to predict the OS level of patients and found that high levels of miR-944 were associated with longer OS in patients [10,67]. However, in this study, using the median truncation method of miR-944 expression levels, we found that there was no significant difference in OS levels between the high and low groups (Appendix A). In conclusion, more clinical data in the future are needed to confirm the prognostic value of miR-944.

In TCGA-BRCA, the expression level of miR-944 in cancer tissues was significantly lower than that in non-cancer tissues, and was significantly correlated with ΔNp63 and TAp63 expression, but not with TP63 CNV. There are multiple subtypes of BrC, and the inconsistency of the abnormal expression pattern of miR-944 in breast cancer may be related to different BrC subtypes. In addition, miR-944 was significantly underexpressed in LUAD tissues and cell lines (such as A549 and H1299), and significantly overexpressed in LUSC tissues (including TCGA-LUSC). Likewise, miR-944 was significantly underexpressed in the tissues and serum of patients with esophageal adenocarcinoma, whereas it was highly expressed in TCGA-ESCA. miR-944 expression was upregulated when cancer cells were derived from highly keratinized cells [17]. During epidermal differentiation, ΔNp63 can upregulate the expression of miR-944 [17], which may be the reason for the difference in the abnormal expression pattern of miR-944 between squamous cell carcinoma and adenocarcinoma.

In CRC, HCC, NPC, and BrC, low levels of miR-944 can serve as a biomarker for poor clinicopathological features and poor prognosis in cancer patients. In CxCa, however, low levels of miR-944 could serve as a biomarker for better clinicopathological features and favorable prognosis in cancer patients. Among the ceRNAs of miR-944, highly expressed circSERPINA3, circHAS2, SNHG6, and PRNCR1 can be used as biomarkers for poor clinicopathological features and poor prognosis in cancer patients. In addition, highly expressed STAT1, a target PCG of miR-944, can serve as a biomarker for poor clinicopathological features and poor prognosis in cancer patients. The above evidence suggests that miR-944 has the potential as a prognostic marker for cancer.

There are still many deficiencies in the research of miR-944. In future studies, the association of miR-944 with resistance to more anticancer drugs needs to be investigated, as well as the relationship between the abnormal expression of miR-944 in cancer and the efficacy of drug treatment.

## 25. Conclusions

This work provides a systematic review of miR-944, points out the potential of miR-944 to become a hot spot in cancer research, and provides potential clues and directions for the follow-up research of miR-944. The bioinformatics analysis herein revealed the potential value of miR-944 as a biomarker for squamous cell carcinoma. At the same time, this paper also points out some controversies and deficiencies in the current research on miR-944. Future research on miR-944 could focus on the molecular regulation of miR-944 dysregulation and its molecular mechanisms associated with antitumor drug resistance and efficacy. This will lay a theoretical foundation for the clinical application of miR-944 in tumors.

## Figures and Tables

**Figure 1 cancers-14-04232-f001:**
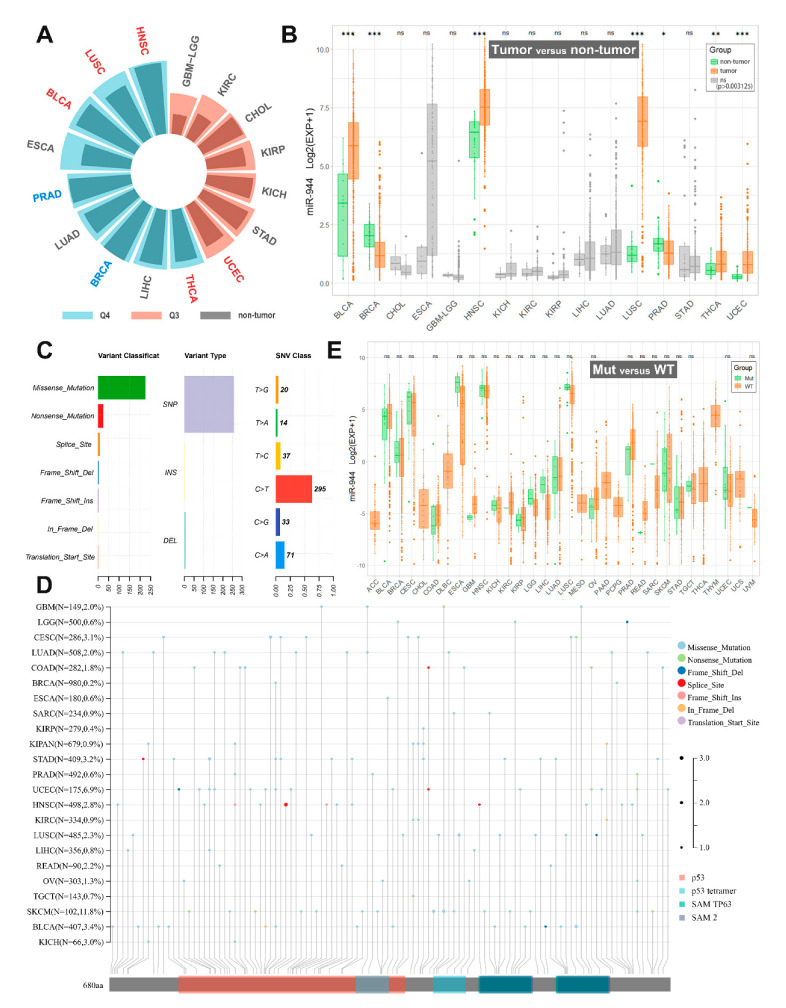
Pan-cancer analysis of miR-944 based on TCGA database. (**A**) Histogram of median quantile expression of miR-944 in non-tumor and tumor groups in the TCGA database. The blue font indicates that miR-944 is significantly low expressed in this cancer type; the red font indicates that miR-944 is significantly highly expressed in this cancer type; (**B**) comparison of miR-944 expression levels between non-tumor and tumor groups in the TCGA database. *** means *p* < 0.0000625; ** means *p* < 0.000625; * means *p* < 0.003125; ns means no significant difference; (**C**) overview of SNVs of TP63; (**D**) mutation types in TP63 protein domains in various cancers; (**E**) differences in the expression level of miR-944 between the TP63 mutant group (Mut) and the wild group (WT). Ns means no significant difference. ACC, adrenocortical carcinoma; BLCA, bladder urothelial carcinoma; BRCA, breast invasive carcinoma; CESC, cervical squamous cell carcinoma and endocervical adenocarcinoma; CHOL, cholangiocarcinoma; COAD, colon adenocarcinoma; DLBC, lymphoid neoplasm diffuse large B-cell lymphoma; ESCA, esophageal carcinoma; GBM, glioblastoma; HNSC, head and neck squamous cell carcinoma; KICH, kidney chromophobe; KIRC, kidney renal clear cell carcinoma; KIRP, kidney renal papillary cell carcinoma; LGG, brain lower grade glioma; LIHC, liver hepatocellular carcinoma; LUAD, lung adenocarcinoma; LUSC, lung squamous cell carcinoma; MESO, mesothelioma; OV, ovarian serous cystadenocarcinoma; PAAD, pancreatic adenocarcinoma; PCPG, pheochromocytoma and paraganglioma; PRAD, prostate adenocarcinoma; READ, rectum adenocarcinoma; SARC, sarcoma; STAD, stomach adenocarcinoma; SKCM, skin cutaneous melanoma; TGCT, testicular germ cell tumor; THCA, thyroid carcinoma; THYM, thymoma; UCEC, uterine corpus endometrial carcinoma; UCS, uterine carcinosarcoma; UVM, uveal melanoma.

**Figure 2 cancers-14-04232-f002:**
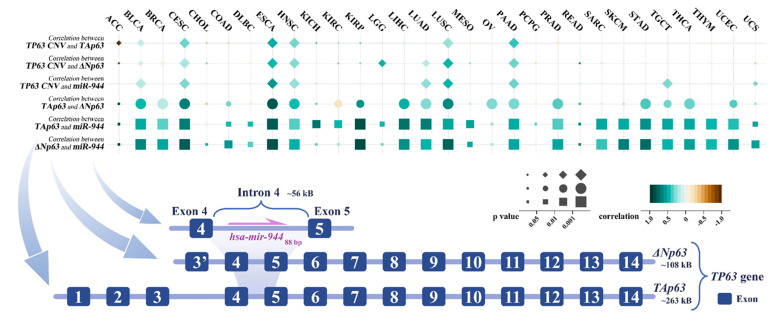
The correlation of miR-944 with TP63 CNV, TAp63, and ΔNp63. The figure indicates the position of hsa-mir-944 in the TP63 gene and shows the correlation of miR-944 with TP63 CNV and the expression of TAp63 and ΔNp63. ACC, adrenocortical carcinoma; BLCA, bladder urothelial carcinoma; BRCA, breast invasive carcinoma; CESC, cervical squamous cell carcinoma and endocervical adenocarcinoma; CHOL, cholangiocarcinoma; COAD, colon adenocarcinoma; DLBC, lymphoid neoplasm diffuse large B-cell lymphoma; ESCA, esophageal carcinoma; HNSC, head and neck squamous cell carcinoma; KICH, kidney chromophobe; KIRC, kidney renal clear cell carcinoma; KIRP, kidney renal papillary cell carcinoma; LGG, brain lower grade glioma; LIHC, liver hepatocellular carcinoma; LUAD, lung adenocarcinoma; LUSC, lung squamous cell carcinoma; MESO, mesothelioma; OV, ovarian serous cystadenocarcinoma; PAAD, pancreatic adenocarcinoma; PCPG, pheochromocytoma and paraganglioma; PRAD, prostate adenocarcinoma; READ, rectum adenocarcinoma; SARC, sarcoma; STAD, stomach adenocarcinoma; SKCM, skin cutaneous melanoma; TGCT, testicular germ cell tumor; THCA, thyroid carcinoma; THYM, thymoma; UCEC, uterine corpus endometrial carcinoma; UCS, uterine carcinosarcoma.

**Figure 3 cancers-14-04232-f003:**
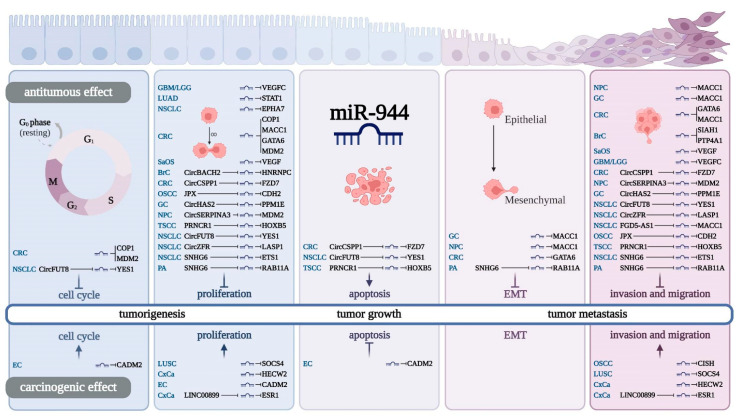
miR-944 and the regulation of cancer cell behaviors. miR-944 can regulate a variety of cancer cell biological behaviors through its ceRNA networks or target genes.

**Figure 4 cancers-14-04232-f004:**
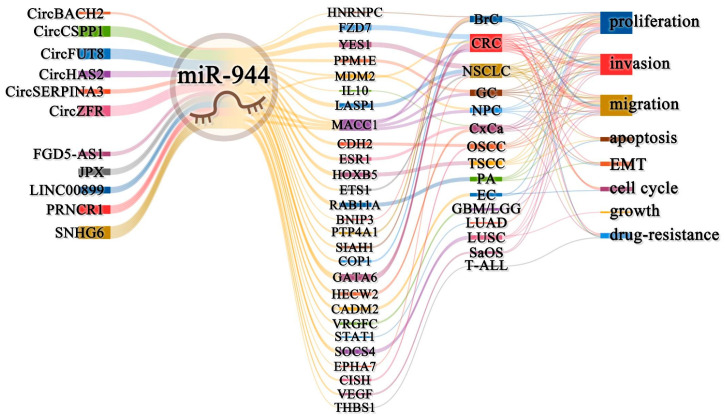
The ceRNA networks of miR-944. The ceRNA networks of miR-944 involve 11 ceRNAs and 27 protein-coding genes. It can regulate a variety of cancer cell biological behaviors, such as proliferation, invasion, migration, apoptosis, drug resistance, EMT, growth, and cell cycle in 16 cancers.

**Figure 5 cancers-14-04232-f005:**
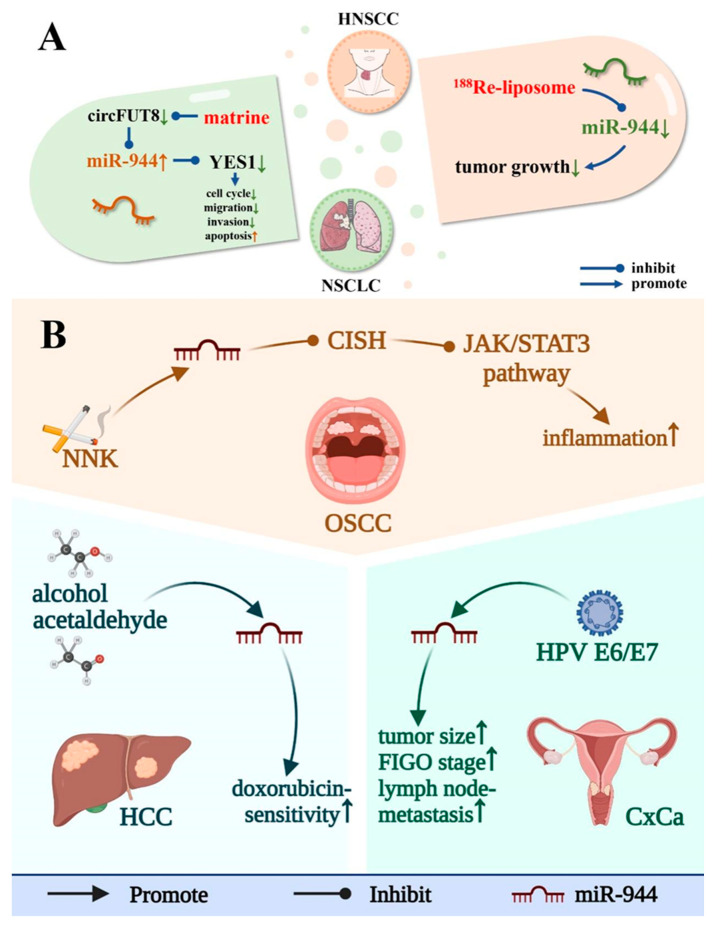
Exogenous factors and drugs affect the level of miR-944. (**A**) miR-944 is involved in the treatment of NSCLC and HNSCC by matrine and ^188^Re-liposome, respectively; (**B**) exogenous factors (including NNK, HPV E6/E7, and alcohol and acetaldehyde) affect the expression level of miR-944 in the OSCC, CxCa, and HCC.

**Figure 6 cancers-14-04232-f006:**
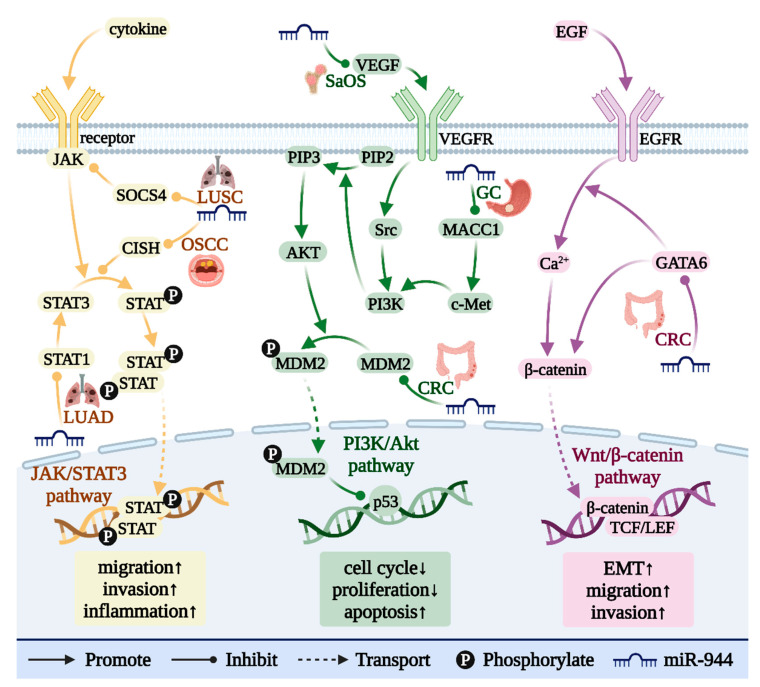
“↑”: activation of the signaling pathway will promote the cell behavior; “↓”: activation of the signaling pathway will inhibit the cell behavior. miR-944 is involved in the regulation of three signaling pathways. miR-944 plays a regulatory role in the occurrence and development of cancer by participating in the Wnt/β-catenin, PI3K/AKT, and Jak/STAT signaling pathways.

**Figure 7 cancers-14-04232-f007:**
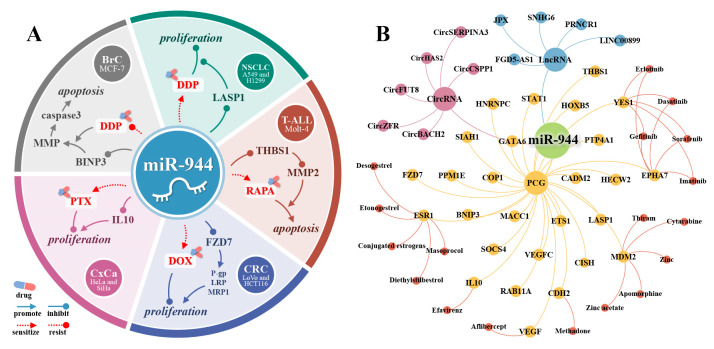
The miR-944-related therapeutic drugs. (**A**) miR-944 is associated with resistance to 4 drugs (DDP, RAPA, DOX, and PTX); (**B**) the target drugs of miR-944′s PCGs and the ceRNA/miR-944/PCG axes in the CADDIE database.

**Table 1 cancers-14-04232-t001:** Aberrant expression of miR-944 in different cancers.

Physiological System	Cancer	miR-944 Expression	Cell Line	Tissue or Serum	Ref.
Nervous system	GBM/LGG	Downregulated	HA1800 versus SHG44, U87MG, and U251MG	Paracancerous tissues versus glioma tissues from 5 patients	[28]
Respiratory system	NPC	Downregulated	NP69 versus C666-1, CNE1, CNE2, and HNE1	Paracancerous tissues versus tumor tissues from 20 NPC patients	[29]
		Downregulated	NP69 versus CNU46, SUNE1, HONE1, 6–10 B, CNE1, CNE2, and HNE1	Nasopharyngeal mucosa tissues from 30 healthy people versus primary tumor tissues from 30 NPC patients	[7]
	LUAD	Downregulated	16HBE versus A549, H1299, SK-Lu-1, and PC-9	Paracancerous tissues versus LUAD tissues from 25 patients	[30]
	LUSC	Upregulated	—	Paracancerous tissues from patients versus SCC tissues from patients	[21]
	NSCLC	Downregulated	BEAS-2B versus H522 and H1975	—	[15]
		Downregulated	BEAS-2B versus H358, H1299, PC-9, and A549	Paracancerous tissues versus tumor tissues from 65 NSCLC patients	[31]
		Downregulated	BEAS-2B versus A549, H226, H292, ANP973, and H1299	Paracancerous tissues versus tumor tissues from 60 NSCLC patients	[32]
		Downregulated	—	Paracancerous tissues versus tumor tissues from 9 NSCLC patients	[2]
Digestive system	TSCC	Downregulated	normal gingival epithelial cells versus SCC-9, CAL-27, and SCC-15	Paracancerous tissues versus TSCC tissues from 57 patients	[33]
	ECa	Downregulated	—	Paracancerous tissues versus adenocarcinoma tissues from 59 eca patients; serum exosomes from healthy persons versus serum exosomes from 59 eca patients	[20]
	GC	Downregulated	GES-1 versus AGS, MKN-1, HGC-27, MKN-45, SGC-7901, and BGC-823	—	[34]
		Downregulated	GES-1 versus SGC-7901, MGC-803, MKN-28, and BGC-823	Paracancerous tissues versus tumor tissues from 40 GC patients	[35]
	HCC	Downregulated	L02 versus Hep3B, Bel-7402, SMMC-7721, Huh7, and SK-HEP-1	Paracancerous tissues versus tumor tissues from 61 HCC patients	[36]
	CRC	Downregulated	HIEC and HEK293 versus HCT116, Caco-2, HT29, SW620, and SW480	—	[9]
		Downregulated	COS7 versus HCT116, LoVo, RKO, HCT15, HT29, SW480, and SW620	—	[37]
		Downregulated	—	Paracancerous tissues versus fresh CRC tissues from 140 CRC patients	[8]
		Downregulated	CCC-HIE-2 versus HT-29, HCT116, SW480, and SW620	Paracancerous tissues versus fresh CRC tissues from 100 CRC patients	[38]
Reproductive system	EC	Upregulated	—	Normal endometrial tissues from 20 non-cancer patients versus tumor tissues from 68 EC patients	[22]
	CxCa	Upregulated	—	Paracancerous tissues versus tumor tissues from 27 cxca patients	[25]
		Upregulated	—	Serum specimens from 24 women with localized disease versus serum specimens from 25 women with metastatic disease	[24]
		Upregulated	HcerEpiC versus HeLa, CaSki, SiHa, and C33A	Paracancerous tissues versus fresh cxca tissues from 70 cxca patients	[23]
		Upregulated	—	50 FFPE normal cervical tissue samples versus 66 FFPE cxca tissue samples	[5]
	BrC	Downregulated	MCF-10A versus MDA-MB-231, MCF-7, MDA-MB-453, ZR-75, and T47-D	Paracancerous tissues versus locally invasive breast tumors tissues from brc patients	[27]
		Upregulated	—	Paracancerous tissues versus tumor tissues from 40 brc patients; serum samples from 30 healthy people versus serum samples from 30 brc patients	[26]
Motor system	SaOS	Downregulated	hFOB1.19 versus MG-63, SAOS-2, HOS, and U2OS	Paracancerous tissues versus tumor tissues from 38 saos patients	[39]
	COF	Downregulated	—	Bone tissues from 10 healthy people versus bone tissues from 9 COF patients	[3]

GBM, glioblastoma; LGG, brain lower grade glioma; NPC, nasopharyngeal carcinoma; LUAD, lung adenocarcinoma; LUSC, lung squamous cell carcinoma; NSCLC, non-small-cell lung cancer; SCC, squamous cell carcinoma; TSCC, tongue squamous cell carcinoma; ECa, esophageal cancer; GC, gastric cancer; HCC, hepatocellular carcinoma; CRC, colorectal cancer; EC, endometrial carcinoma; CxCa, cervical cancer; BrC, breast cancer; SaOS, osteosarcoma; COF, cemento-ossifying fibroma.

**Table 2 cancers-14-04232-t002:** Comparison of miR-944 in TCGA dataset with existing data.

TCGA Cancers	Sample Size (T/N)	miR-944 Expression in TCGA	miR-944 Expression in the Present Studies
BLCA	405/18	Upregulated; Q4	Not studied
BRCA	624/74	Downregulated; Q4	Downregulated in BrC tissues and BrC cells (MDA-MB-231, MCF-7, MDA-MB-453, ZR-75, and T47-D) [27]; and Upregulated in BrC tissues and serum sample of BrC patients [26]
CHOL	20/8	ns; Q3	Not studied
ESCA	176/8	ns; Q4	Downregulated in ECa tissues and serums of ECa patients [20]
GBM/LGG	209/3	ns; Q3	Downregulated in GBM/LGG tissues and GBM/LGG cells (SHG44, U87MG, and U251MG) [28]
HNSC	485/44	Upregulated; Q4	Not studied
KICH	49/8	ns; Q3	Not studied
KIRC	108/19	ns; Q3	Not studied
KIRP	155/23	ns; Q3	Not studied
LIHC	324/47	ns; Q4	Downregulated in HCC tissues and HCC cells (Hep3B, Bel-7402, SMMC-7721, Huh7, and SK-HEP-1) [36]
LUAD	430/40	ns; Q4	Downregulated in LUAD tissues and LUAD cells (A549, H1299, SK-Lu-1, and PC-9) [30]
LUSC	334/44	Upregulated; Q4	Upregulated in LUSC tissues [21]
PRAD	437/50	Downregulated; Q4	Not studied
STAD	303/26	ns; Q3	Downregulated in GC tissues and GC cells (AGS, MKN-1, HGC-27, MKN-45, SGC-7901, MGC-803, BGC-823, and MKN-28) [34,35]
THCA	420/50	Upregulated; Q4	Not studied
UCEC	330/26	Upregulated; Q3	Upregulated in EC tissues [22]

Q3, 0.5–0.75 quantile; Q4, 0.75–1.0 quantile; BLCA, bladder urothelial carcinoma; BRCA, breast invasive carcinoma; CHOL, cholangiocarcinoma; COAD, colon adenocarcinoma; ESCA, esophageal carcinoma; GBM, glioblastoma; HNSC, head and neck squamous cell carcinoma; KICH, kidney chromophobe; KIRC, kidney renal clear cell carcinoma; KIRP, kidney renal papillary cell carcinoma; LGG, brain lower grade glioma; LIHC, liver hepatocellular carcinoma; LUAD, lung adenocarcinoma; LUSC, lung squamous cell carcinoma; PRAD, prostate adenocarcinoma; STAD, stomach adenocarcinoma; THCA, thyroid carcinoma; UCEC, uterine corpus endometrial carcinoma.

**Table 3 cancers-14-04232-t003:** The target genes of miR-944 and cell behaviors.

Cancer	PCG	Effect in Vitro	Cell Line	Effect in Vivo	Xenograft Model	Ref.
BrC	BNIP3	DDP-resistance↑	MCF-7	—	—	[26]
	HNRNPC	Proliferation↓	MCF-7 and MDA-MB-231	—	—	[10]
	PTP4A1 and SIAH1	Invasion↓ and migration↓	MDA-MB-231	—	—	[27]
CRC	COP1 and MDM2	Cell cycle↓ and proliferation↓	HCT116	Tumor growth↓	HCT116 cell xenograft in BALB/c nude mice	[37]
	FZD7	Proliferation↓, invasion↓, migration↓, apoptosis↑, and DOX-resistance↓	LoVo and HCT116	Tumor growth↓	LoVo cell xenograft in nude mice	[13]
	GATA6	Invasion↓, migration↓, and EMT↓	SW480 and HCT116	—	—	[8]
		Proliferation↓, invasion↓, and migration↓	SW480 and HCT116	—	—	[38]
	MACC1	Proliferation↓, invasion↓, and migration↓	SW620	—	—	[9]
CxCa	ESR1	Proliferation↑, invasion↑, and migration↑	CaSki and SiHa	—	—	[23]
	HECW2	Proliferation↑, invasion↑, and migration↑	HeLa	—	—	[25]
	IL10	PTX-resistance↓	HeLa and SiHa	—	—	[14]
EC	CADM2	Cell cycle↑, proliferation↑, and apoptosis↓	Ishikawa and KLE	Tumor growth↑	Ishikawa cell xenograft in BALB/c nude mice	[22]
GBM/LGG	VEGFC	Proliferation↓ and migration↓	HUVECs	Tumor growth↓ and angiogenesis↓	SHG44 cell xenograft in nude mice	[28]
GC	MACC1	Invasion↓, migration↓, and EMT↓	GES-1 and MGC-803	—	—	[35]
	PPM1E	Proliferation↓, invasion↓, and migration↓	HGC-27 and MKN-45	—	—	[34]
LUAD	STAT1	Proliferation↓	A549 and H1299	Tumor growth↓	A549 cell xenograft in BALB/c nude mice	[30]
LUSC	SOCS4	Growth↑, proliferation↑, invasion↑, and migration↑	CALU-1 and H520	—	—	[21]
NPC	MACC1	Invasion↓, migration↓, and EMT↓	6–10B and C666-1	—	—	[29]
	MDM2	Proliferation↓ and invasion↓	HONE-1	—	—	[7]
NSCLC	LASP1	Proliferation↓, invasion↓, migration↓, and DDP-resistance↓	A549 and H1299	Tumor growth↓ and DDP-resistance↓	A549 cell xenograft in BALB/c nude mice	[11]
	MACC1	Invasion↓ and migration↓	A549 and H1299	—	—	[31]
	EPHA7	Proliferation↓	EPLC-32M1, A549, and XLA-07	—	—	[42]
	ETS1	Proliferation↓ and migration↓	A549	—	—	[32]
	YES1	Cell cycle↓, proliferation↓, invasion↓, migration↓, and apoptosis↑	H522 and H1975	Tumor growth↓	H522 cell xenograft in nude mice	[15]
OSCC	CDH2	Proliferation↓, invasion↓, and migration↓	SCC-15 and SCC-9	—	—	[43]
	CISH	Invasion↑ and migration↑	OEC-M1 and SCC-25	—	—	[44]
PA	RAB11A	Proliferation↓, invasion↓, migration↓, and EMT↓	HP75	—	—	[45]
SaOS	VEGF	Proliferation↓ and invasion↓	MG-63 and U2OS	—	—	[39]
T-ALL	THBS1	RAPA-resistance↓	Molt-4	—	—	[12]
TSCC	HOXB5	proliferation↓, invasion↓, migration↓, and apoptosis↑	SCC-9 and CAL-27	Tumor growth↓	SCC-9 cell xenograft in nude mice	[33]

“↓” means that the biological behavior is inhibited, “↑” means that the biological behavior is promoted; BrC, breast cancer; CRC, colorectal cancer; CxCa, cervical cancer; EC, endometrial carcinoma; GBM, glioblastoma; LGG, brain lower grade glioma; GC, gastric cancer; HCC, hepatocellular carcinoma; GC, gastric cancer; HCC, hepatocellular carcinoma; LUAD, lung adenocarcinoma; LUSC, lung squamous cell carcinoma; NPC, nasopharyngeal carcinoma; NSCLC, non-small-cell lung cancer; OSCC, oral squamous cell carcinoma; PA, pituitary adenoma; SaOS, osteosarcoma; T-ALL, T-cell acute lymphoblastic leukemia; TSCC, tongue squamous cell carcinoma; DDP, cisplatin; DOX, doxorubicin; PTX, Paclitaxel; RAPA, rapamycin; EMT, epithelial mesenchymal transition.

**Table 4 cancers-14-04232-t004:** The ceRNAs of miR-944.

CeRNA Axis	Cancer	Binding Site of ceRNA and miR-944		Binding Site of miR-944 and PCG		Ref.
		ceRNA (5’-…-3’)	miR-944 (3’-…-5’)	PCG (5’-…-3’)	miRNA (3’-…-5’)	
CircBACH2/miR-944/HNRNPC	BrC	—	—	—	—	[10]
CircCSPP1/miR-944/FZD7	CRC	CAAUAAUUU	GUUAUUAAA	UAAUUU	AUUAAA	[13]
CircFUT8/miR-944/YES1	NSCLC	AcCaGAGAgAAUAAUU	UaGgCUaCaUgUUAUUAA	GAUuUcaAAUAAUU	CUAcAugUUAUUAA	[15]
CircHAS2/miR-944/PPM1E	GC	AgAATAATT	UgUUAUUAA	AAUAAUU	UUAUUAA	[34]
CircSERPINA3/miR-944/MDM2	NPC	GUUUCAACA	CAAAGUUGU	uCUuCucUuUAguAUAAUU	gAGuAgGcuAcAUguUAUUAA	[7]
CircZFR/miR-944/IL10	CxCa	CAAUAAUU	GUUAUUAA	AUAAUU	UAUUAA	[14]
CircZFR/miR-944/LASP1	NSCLC	CAAUAAUU	GUUAUUAA	AUAAUU	UAUUAA	[11]
FGD5-AS1/miR-944/MACC1	NSCLC	AUGUACuAAUAAUUU	UACAUGUUAUUAAA	AUAAUU	UAUUAA	[31]
JPX/miR-944/CDH2	OSCC	AUcGgAgAAUAAUU	UACaUgUUAUUAA	AUAAUU	UAUUAA	[43]
LINC00899/miR-944/ESR1	CxCa	AUuCugUuUACagaAAUAAUU	UAgGcuAcAUGUUAUUAA	AUAAUU	UAUUAA	[23]
PRNCR1/miR-944/HOXB5	TSCC	AAUAAUU	UUAUUAA	AAUAAUU	UUAUUAA	[33]
SNHG6/miR-944/ETS1	NSCLC	UuuGAaGAAAUAAUUU	AggCUaCaUgUUAUUAAA	ucCAUgaGAUuUgAAUAgAUUU	gaGUAggCUAcAugUUAUUAAA	[32]
SNHG6/miR-944/RAB11A	PA	UuuGAaGAAAUAAUU	AggCUaCaUgUUAUUAA	CAAUAAUU	GUUAUUAA	[45]
miR-944/BNIP3	BrC	—	—	AAUAAUUU	UUAUUAAA	[26]
miR-944/CADM2	EC	—	—	AAUAAUU	UUAUUAA	[22]
miR-944/CISH	OSCC	—	—	UuCAUGaaAuAAUAAUU	AgGcUACaUgUUAUUAA	[44]
miR-944/COP1	CRC	—	—	UuGaaUAaAaAUAAAU	GgCuaCAuGuUAUUAA	[37]
miR-944/EPHA7	NSCLC	—	—	AAUAAUU	UUAUUAA	[42]
miR-944/GATA6	CRC	—	—	AAUAAUUU	UUAUUAAA	[38]
miR-944/HECW2	CxCa	—	—	CUgugCaucUaaguAAUAAUUU	GAguaGgcuAcaugUUAUUAAA	[25]
miR-944/MACC1	CRC	—	—	AAUAAUU	UUAUUAA	[9]
	GC	—	—	AAUAAUU	UUAUUAA	[35]
	NPC	—	—	AAUAAUU	UUAUUAA	[29]
miR-944/MDM2	CRC	—	—	UaaUUuUaAAUAAUU	GcuACaUgUUAUUAA	[37]
miR-944/PTP4A1	BrC	—	—	AAUAAUUU	UUAUUAAA	[27]
miR-944/SIAH1	BrC	—	—	AAUAAUU	UUAUUAA	[27]
miR-944/SOCS4	LUSC	—	—	GAUccaAAUAAUU	CUAcaugUUAUUAA	[21]
miR-944/STAT1	LUAD	—	—	UAUcCaAaGcugAAUAcAUU	GUAgGcUaCaugUUAUUAA	[30]
miR-944/THBS1	T-ALL	—	—	—	—	[12]
miR-944/VEGF	SaOS	—	—	AAUAAUU	UUAUUAA	[39]
miR-944/VEGFC	GBM/LGG	—	—	AAUAAUU	UUAUUAA	[28]

BrC, breast cancer; CRC, colorectal cancer; NSCLC, non-small-cell lung cancer; GC, gastric cancer; NPC, nasopharyngeal carcinoma; CxCa, cervical cancer; OSCC, oral squamous cell carcinoma; TSCC, tongue squamous cell carcinoma; PA, pituitary adenoma; EC, endometrial carcinoma; HCC, hepatocellular carcinoma; LUSC, lung squamous cell carcinoma; LUAD, lung adenocarcinoma; T-ALL, T-cell acute lymphoblastic leukemia; SaOS, osteosarcoma; GBM, glioblastoma; LGG, brain lower grade glioma. Contrary to the above, miR-944 is associated with poorer prognosis in HNSCC, EC, and CxCa. In HNSCC, high levels of miR-944 were associated with shorter OS [16]. In EC, high levels of miR-944 were associated with later FIGO stage and poor pathological classification, and shorter OS [22]. In CxCa, high levels of miR-944 were associated with larger tumor size, later FIGO stage, more lymph node metastasis, and shorter OS [5].

**Table 5 cancers-14-04232-t005:** Prognostic values of miR-944.

Cancer	Sample Size	miR-944 Expression	Clinicopathological Characteristics	Prognostic Values of miR-944 Overexpression	Ref.
BrC	1061	Downregulated	—	Longer OS	[67]
	1062	Downregulated	Earlier clinical stages and TNM stage	Longer OS	[10]
CRC	86	Downregulated	Earlier tumor stage, Earlier TNM stage, less lymph node metastasis and distant metastasis	Longer OS and PFS	[9]
	265	Downregulated	—	Longer OS	[37]
	140	Downregulated	Earlier TNM stage, small lymph node status, and less liver metastasis	Longer OS	[8]
CxCa	66	Upregulated	Advanced clinical stages	Shorter OS	[5]
EC	68	Upregulated	Advanced FIGO stages and poorer pathology classification	Shorter OS	[22]
HNSCC	522	Upregulated	—	Shorter OS	[16]
NPC	30	Downregulated	Earlier clinical stage	Longer OS	[7]

BrC, breast cancer; CRC, colorectal cancer; CxCa, cervical cancer; EC, endometrial carcinoma; HCC, hepatocellular carcinoma; HNSCC, head and neck squamous cell carcinoma; NPC, nasopharyngeal carcinoma; TNM, tumor node metastasis; FIGO, international federation of gynecology and obstetrics; OS, overall survival; DFS, disease-free survival; PFS, progression-free survival.

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
