# Peer review of "Novel Insights into miR-944 in Cancer"

_cancers, 2022, doi:10.3390/cancers14174232_

Round 1
Reviewer 1 Report
Excellent review written by the Jinze Shen et all.
I really appreciate the authors, the way they have applied the various bioinformatics and illustration tools to depict the data associated with miRNA944 with pan cancer conditions.
I have only few of concerns:
1) Figure 1 which is not that much clear, authors should improve the quality of fig. 1.
2) Authors should connect the expression level of miRNA944 along with the overall survival curve also.
3) Kindly discuss, "What would be the gender and ethnicity based expression disparity ratio of miRNA944?"
4) The linkage of life style impact over the miRNA 944 expression will enhance the quality of this review.
5) One graphical abstract will potentiate the gravity of this review.
Reviewer 2 Report
Manuscript by Shen et al presents a review on “Novel insights into miR-944 in cancer”. The paper systematically reviewed related publications about association between miR-944 aberrations and its significance in various cancers.
The merit of the paper is their comprehensive collection of miR-944 relevant literature and detailed bioinformatic analysis, which will facilitate the readers’ understanding of miR-944’s role and regulation in oncogenesis. However, it could be strengthened if the authors could integrate more discussions regarding paradox functions of miR-944 in different cancer or reports. The discussion section of the current version was a good summary of the facts from reported literature, rather than integrated discussion.
Presentation of Table 5 is problematic. The “prognostic value” in the table and the content in the text of several cancers are inconsistent. For example, the text says “In BrC, low expression of miR-944 was associated with advanced clinical-stage, late TNM stage, and shorter OS”. But the table indicate “Longer OS”. This is contradictive and confusing. If the authors want to say “because lower expression is associated with shorter OS, then higher expression would suggest longer OS”, then it has to meet two conditions: 1. Title of the table should be “Prognostic values of miR-944 overexpression”; 2. There are reports showing miR-944 overexpression exist in breast cancers. The question is whether miR-944 overexpression is detected in breast cancer. According to the text, low expression of miR-944 in breast cancer was associated with advanced clinical-stage, late TNM stage, and shorter OS. This is a critical question that has to be clarified.
